# Synthetic Approaches to Zetekitoxin AB, a Potent Voltage-Gated Sodium Channel Inhibitor

**DOI:** 10.3390/md18010024

**Published:** 2019-12-26

**Authors:** Kanna Adachi, Hayate Ishizuka, Minami Odagi, Kazuo Nagasawa

**Affiliations:** Department of Biotechnology and Life Science, Tokyo University of Agriculture and Technology (TUAT), 2-24-16, Naka-cho, Koganei city, Tokyo 184-8588, Japan; s175364y@st.go.tuat.ac.jp (K.A.); s189783w@st.go.tuat.ac.jp (H.I.)

**Keywords:** saxitoxin, zetekitoxin AB, voltage-gated sodium channel, guanidine alkaloid

## Abstract

Voltage-gated sodium channels (Na_V_s) are membrane proteins that are involved in the generation and propagation of action potentials in neurons. Recently, the structure of a complex made of a tetrodotoxin-sensitive (TTX-s) Na_V_ subtype with saxitoxin (STX), a shellfish toxin, was determined. STX potently inhibits TTX-s Na_V_, and is used as a biological tool to investigate the function of Na_V_s. More than 50 analogs of STX have been isolated from nature. Among them, zetekitoxin AB (ZTX) has a distinctive chemical structure, and is the most potent inhibitor of Na_V_s, including tetrodotoxin-resistant (TTX-r) Na_V_. Despite intensive synthetic studies, total synthesis of ZTX has not yet been achieved. Here, we review recent efforts directed toward the total synthesis of ZTX, including syntheses of 11-saxitoxinethanoic acid (SEA), which is considered a useful synthetic model for ZTX, since it contains a key carbon–carbon bond at the C11 position.

## 1. Introduction

### 1.1. Voltage-Gated Sodium Channel Isoforms 

Voltage-gated sodium channels (Na_V_s) are membrane proteins involved in neuronal excitation and transmission [1]. Ten subtypes, Na_V_1.1–1.9 and Na_V_X, have been identified based on sequence determination (Table 1) [2]. These subtypes can be grouped into two types depending upon their sensitivity to the pufferfish toxin, tetrodotoxin (TTX) [3,4,5,6,7]: tetrodotoxin-sensitive Na_V_s (TTX-s Na_V_s 1.1–1.4, 1.6, and 1.7) are significantly inhibited by TTX, while tetrodotoxin-resistant Na_V_s (TTX-r Na_V_s 1.5, 1.8, 1.9) are not [8,9,10,11,12,13,14,15,16,17,18,19,20,21,22,23]. Subtype-selective modulators of Na_V_s are required for studies to establish the biological functions of these subtypes. Some of the subtypes are also considered to be potential drug targets; for example, Na_V_1.7 and 1.8 are potential targets for pain treatment [24,25,26,27,28,29]. Therefore, there is great interest in the development of drugs targeting specific subtypes [30,31].

### 1.2. Saxitoxin As A Na_V_ Modulator

Saxitoxin (STX, **1**) is a guanidine alkaloid with potent and specific inhibitory activity towards Na_V_s (Figure 1) [32,33]. It has long been known as a shellfish toxin. In 1937, Sommer and co-workers found that toxin-free bivalves, including the dinoflagellate *Gonyaulax catenella*, became poisoned in seawater, and they revealed that the real producer of STX (**1**) is algae [34,35]. Then, STX (**1**) was first isolated from Alaska butter clams by Schantz’s group in 1957 [36,37]. Rapport’s group subsequently isolated the same toxin from the same shellfish, and named it saxitoxin [38]. Structural elucidation was troublesome. Initially, tri- or tetracyclic structures were proposed based upon the molecular formula and the presence of two guanidines as functional groups. Finally, the structure of STX (**1**) was independently determined by the two groups by means of X-ray analysis in 1975 [39,40]. STX (**1**) consists of ten carbons, seven nitrogens, and four oxygens, and all the carbons except for C11 are connected with heteroatoms. STX (**1**) contains five- and six-membered cyclic guanidines, which have different pKa values of 8.7 and 12.4, respectively; the five-membered one is less basic, presumably due to its less planar structure [41].

STX (**1**) binds to the pore-forming region of the alpha-loop of Na_V_ and blocks the influx of sodium cation in a similar manner to tetrodotoxin (TTX) [42,43]. Recently, Yan’s group determined the X-ray structure of the complexes of STX (**1**) with Na_V_Pas derived from American cockroach and human-derived Na_V_1.7 by using cryoEM (Figure 2) [44,45]. They found that the carbamoyl group at C13 in STX (**1**) interacts with Gly1407 and Thr1409 in domain III, the two guanidines interact with Glu364 in domain I and Glu930 in domain II, and the geminal diol interacts with Asp1701 in domain IV. Interestingly, residues 1409 and 1410, located in the P2 loop of domain III in Na_V_1.7, were mutated to Thr and Ile from Met and Asp, respectively, which may explain why STX (**1**) has a weaker affinity for Na_V_1.7 compared with other subtypes (Met1409 and Asp1410 are conserved in the other subtypes of TTX-s) [46,47].

### 1.3. Natural Analogs of Saxitoxin, Including Zetekitoxin AB (ZTX)

To date, more than 50 kinds of natural analogs of saxitoxin have been reported, of which most are modified at N1 (R^1^), C11 (i.e., R^2^ and R^3^), or C13 (R^4^) in the common structure shown in Figure 3A [48]. For example, neosaxitoxin (neoSTX, **2**) is hydroxylated at N1, decarbamoylsaxitoxin (dcSTX, **3**) has a hydroxyl group at C13, and gonyautoxins I–III (GTX I–III, **5**–**7**, respectively) have a sulfate ester at C11; all of these analogs show similar Na_V_-inhibitory activity to STX (**1**). 

Among the STX (**1**) derivatives, zetekitoxin AB (ZTX, **8**) has an unusual structure [49]. ZTX (**8**) was isolated from skin of the Panamanian dart-poison frog *Atelopus zeteki* in 1969 by Mosher and co-workers [50,51]. It has extremely potent Na_V_-inhibitory activity (more than 600-fold greater than that of STX (**1**)), with IC_50_ values of 6.1 pM, 65 pM, and 280 pM for Na_V_1.2, Na_V_1.4, and TTX-r subtype Na_V_1.5, respectively [49]. Thus, there is great interest in the mode of action of ZTX (**8**), but studies are hampered by the fact that *Atelopus zeteki* is designated as an endangered species. Therefore, a chemical synthesis of ZTX (**8**) is needed. However, ZTX (**8**) contains a macrocyclic lactam structure in which isoxazolidine is bridged from C6 to C11, and an *N*-hydroxycarbamate is linked via a methylene group at N7 [49]. These structural features make ZTX (**8**) synthetically challenging. So far, several synthetic approaches have been reported, but a total synthesis of **8** has not yet been achieved.

### 1.4. Scope of This Review 

Synthetic studies of STX (**1**) and its analogs have been extensive, and several total syntheses have been achieved [52,53,54,55,56,57,58,59,60,61], as recently reviewed by Du Bois [62]. Approaches for developing of subtype-selective modulators based on the STX structure have also been explored [25,47,63,64,65]. However, in this review, we focus on recent synthetic work related to ZTX (**8**). As described above, ZTX (**8**) has a characteristic macrolactam structure though C6 to C11 with an isoxazolidine ring system, and is structurally quite distinct from other STX analogs. To achieve total synthesis of ZTX (**8**), two key issues must be addressed: (i) carbon–carbon bond formation at the C11 position in the STX skeleton, and (ii) macrolactam formation of the carboxylic acid at C6 with isoxazolidine nitrogen (Figure 4). Regarding the first issue, the STX derivative 11-saxitoxinethanoic acid (SEA, **9**) has been used as a synthetic model for **8**, since it also has a carbon–carbon bond at the C11 position. As for the second issue, stereoselective synthesis of disubstituted isoxazolidine and oxidation to carboxylic acid at C13, followed by amide formation with the isoxazolidine, have been examined. First, we will consider recent progress in the total synthesis of SEA (**9**).

## 2. Development of Carbon–Carbon Linkage at C11 of STX, And Application to The Synthesis of 11-Saxitoxinethanoic Acid (SEA, 9)

The STX analog 11-saxitoxinethanoic acid (SEA, **9**) was isolated from *Atergatis floridus*, an Indo-Pacific crab from the family Xanthidae, by Onoue and co-workers (Figure 5) [66]. SEA (**9**) has an acetic acid moiety linked to C11 through a carbon–carbon bond, as seen in ZTX (**8**), and is regarded as a promising synthetic model compound for **8** in terms of construction of the carbon–carbon connection at C11.

Recently, three total synthesis of SEA (**9**) were independently reported, including one by our group [67,68,69]. When **9** was first isolated, its toxicity to mice was reported to be 830 μmol/MU, which is similar to that of gonyautoxin II (GTX II, **6**) and one-third of that of STX (**1**), but no information about the Na_V_-inhibitory activity was provided. After the synthesis of **9**, Du Bois and our group independently evaluated the Na_V_-inhibitory activity of **9**. Nagasawa, Yotsu-Yamashita, and co-workers evaluated the Na_V_-inhibitory activity of SEA (**9**) by utilizing neuroblastoma Neuro 2A cells, which is known to express Na_V_1.2, 1.3, 1.4, and 1.7 [70], and found moderate inhibitory activity with an IC_50_ value of 47.0 ± 1.2 nM (Figure 6B) [67]. Du Bois and co-workers evaluated the inhibitory activity of **9** against Na_V_1.4, and found that SEA (**9**) showed similar inhibitory activity to gonyautoxin III (GTX III, **7**) (**9**: IC_50_ = 17 ± 1.9 nM; GTX III (**7**): IC_50_ = 14.9 ± 2.1 nM), even though it was a diastereomeric mixture of α:β = 3:1 at C11 (Figure 6A) [68]. They suggested that the β-form of **9** binds to Na_V_ preferentially, and then the α-form of **9** isomerizes to the β-form, which shows a similar level of inhibitory activity to GTX III (**7**) (Figure 6C).

### 2.1. Carbon–Carbon Bond Formation at C11 by Mukaiyama Aldol Condensation Reaction, as Applied for The Synthesis of (+)-SEA by Nagasawa’s Group

For the construction of a carbon–carbon bond at C11, Nagasawa and co-workers utilized ketone **10**, which was previously developed by their group [67], to install an acetic acid equivalent at C11. They firstly investigated the alkylation reaction of the enolate of ketone **10a** with alpha-halo-ethyl acetate. With various bases and halogens, the alkylation did not take place at all, and the starting ketone **10a** was recovered. Next, they investigated the Mukaiyama aldol reaction [71,72,73]. Thus, silyl enol ethers **11a** and **11b** were synthesized from the ketone by reaction with *tert*-butyldimethylsilyl chloride in the presence of NaHMDS as a base. Then, the Mukaiyama aldol reaction was examined with ethyl glyoxylate under various conditions. Lewis acids, such as TiCl_4_ or BF_3_ Et_2_O [74,75], removed the *tert*-butoxycarbonyl (Boc) protecting group of guanidine, and no coupling products with ethyl glyoxylate were obtained. In the case of the fluoride anion agent Bu_4_NF [76], the reaction did not proceed at all. On the other hand, with anhydrous tetrabutyl bisfluorotriphenylphosphine stannate, developed by Raimundo and co-workers [77], the coupling reaction with ethyl glyoxylate proceeded very well to afford the aldol-condensation product **12a** a 96% yield (Table 2). Aromatic aldehydes were tolerated, as well as aliphatic aldehydes, and the corresponding aldol condensation products **12a–i** were obtained with 42%–80% yield. This reaction afforded mixtures of regioisomers in ratios of 5:1 to >10:1.

With the aldol condensation product **12b** in hand, Nagasawa and co-workers went on to achieve a total synthesis of (+)-SEA (**9**) for the first time (Scheme 1). Thus, selective reduction of the enone moiety in **12b** was carried out with L-selectride, and the protecting group of *tert*-butyldimethylsilyl (TBS) ether was removed with triethylamine trihydrofluoride (3HF-TEA). The resulting alcohol was reacted with trichloroisocyanate, followed by hydrolysis of the trichloroacetyl group with triethylamine in methanol to give carbamoyl **15**. After hydrolysis of ethyl ester in **15** with lithium hydroxide, the Boc group was removed with TFA to give (+)-SEA (**9**). 

### 2.2. Carbon–Carbon Bond Formation At C11 by Stille Coupling Reaction, As Applied for The Synthesis of (+)-SEA by Du Bois’ Group

Another approach for the construction of the carbon–carbon bond at C11 in STX was explored by Du Bois and co-workers, who employed Stille coupling reaction conditions [68]. They firstly examined the coupling reaction of zinc enolate of ethyl acetate or the stannane enolate of ethyl acetate-type agents with vinyl halide **17**, which was prepared from **20**, developed by their group (Scheme 2), in the presence of palladium catalyst (Table 3, entries 1 and 2) [78,79,80,81,82,83]. Under the conditions examined, decomposition of the starting substrate was observed in the case of zinc agent, and no reaction occurred with the stannane agent. Then they examined the Stille coupling reaction, using vinyl stannane for the construction of the carbon–carbon bond at C11 [84]. A Stille coupling reaction of vinyl iodide **17** with tributyl(vinyl)tin was examined in the presence of a catalytic amount of Pd(PPh_3_)_4_, with CuI as an additive (a standard condition). Unfortunately, only a trace amount of the corresponding coupling product of **18c** was obtained (entry 3). Then, they changed vinyl stannane to cis-tributyl (2-ethoxyvinyl) tin, and included LiCl as an additional additive. Under these conditions, the corresponding coupling product **18d** was obtained with 67% yield (entry 4) [85,86]. Interestingly, poor reproducibility or low yield of the coupling reaction was observed when they used a highly oxidized vinyl stannane agent, tributyl(2,2-diethoxyvinyl)stannane (entry 5). This issue was successfully overcome by switching from CuI to copper(I) thiophene-2-carboxylate (CuTC), and **19** was obtained with 60% yield and with good reproducibility (entry 6) [87]. 

Based upon the Stille coupling strategy, Du Bois and co-workers achieved a total synthesis of SEA (**9**), as shown in Scheme 2, including the synthesis of vinyl halide **17** as a substrate for the Stille coupling reaction. Firstly, vinyl halide **17** was synthesized from **20** via Mislow–Evans [2,3] rearrangement: bisguanidine **20** was converted to *N*,*S*-acetal **21** by reaction with benzenethiol in the presence of BF_3_·Et_2_O with 84% yield. Upon treatment of **21** with urea–hydrogen peroxide (UHP), the Mislow–Evans [2,3] rearrangement reaction [88,89] took place under heating in the presence of sodium benzenthiolate, and allylic alcohol **23** was obtained with 81% yield in two steps. After oxidation of the alcohol with Dess–Martin periodinate, the resulting enone **24** was reacted with iodine in the presence of pyridine to give vinyl iodide **17 [90,91]**, which was further elaborated to **19** by Stille coupling reaction with **25** with 60% yield. Then, the double bond in enone **19** was hydrogenated under high pressure in the presence of Crabtree catalyst **26**. Deprotection of the *tert*-butyldiphenylchlorosilane (TBDPS) ether in **27** with tetrabutylammonium (TBAF) was followed by installation of a carbamoyl group on the resulting hydroxyl group. Finally, deprotection of Tces and Troc and hydrolysis of the ester group were carried out to give (+)-SEA (**9**).

### 2.3. Carbon–Carbon Bond Formation at C11 by C-Alkylation, As Applied for The Synthesis of (+)-SEA by Looper’s Group

In 2019, Looper and co-workers successfully constructed a carbon–carbon bond at C11 in STX, and reported a total synthesis of SEA (**9**) [69]. They initially examined C-alkylation with ketone **28** and electrophiles in the presence of variety of bases, such as lithium bis(trimethylsilyi)amide (LHMDS), lithium diisopropyl amide (LDA), potassium bis(trimethylsilyi)amide (KHMDS) and sodium bis(trimethylsilyi)amide (NaHMDS). In addition, they examined various electrophiles (haloacetates, allylic halides, and propargylic halides), but no reaction took place, as Nagasawa and co-workers had found (Scheme 3) [67]. 

On the other hand, they found that C-alkylation took place upon reaction of ketone **28** and *tert*-butyl bromoacetate via the generation of zinc enolate by reaction with LiHMDS in the presence of Et_2_Zn, affording a mixture of **29a** and its Boc-deprotected derivative **29b** with 60% yield (based on the starting material) (Scheme 4). By means of this alkylation strategy, they succeeded in synthesizing ZTX (**9**) as follows. Deprotection of TBDPS ether in **29** with TBAF followed by carbamoylation of the resulting alcohol resulted in **30**. Finally, total synthesis of SEA (**9**) was achieved by reaction with TFA to hydrolyze the ester and deprotect the Boc and DPM groups. 

### 2.4. Na_V_-Inhibitory Activity of Synthesized, C11-Substituted Saxitoxin Analogs

Based on the method described above for constructing a carbon–carbon bond at C11 in STX, Nagasawa and co-workers synthesized a series of STX analogs bearing substituents at C11, and evaluated the Na_V_-inhibitory activity of these analogs at the cellular level [67]. 

Beside SEA (**9**), they synthesized dicarbamoyl SEA (dcSEA, **31**), 11-saxitoxin ethyl ethanoate (SEE, **32**), and 11-benzylidene STX (**33a**), and evaluated their Na_V_-inhibitory activity in mouse neuroblastoma Neuro 2A cells, which is known to express Na_V_1.2, 1.3, 1.4, and 1.7 [70]. SEA (**9**) showed potent inhibitory activity with an IC_50_ value of 47 ± 12 nM, which is twice as potent as decarbamoyl saxitoxin (dcSTX (**3**), IC_50_ = 89 ± 36 M) (Figure 7, Table 4). The dcSEA (**31**) and SEE (**32**) showed IC_50_ values of 5700 ± 3.1 and 185 ± 74 nM, respectively. Interestingly, 11-benzylidene STX (**33a**) was a potent inhibitor, with an IC_50_ value of 16.0 ± 6.9 nM. Although the inhibition mode of **33a** has not been clarified yet, the non-hydrated keto group at C12 in **33** might bind efficiently with Na_V_, resulting in potent inhibitory activity.

Next, they further synthesized 11-substituted STX analogs **33b**–**f**, and elucidated their subtype selectivity towards Na_V_1.2, 1.5, and 1.7, using the whole-cell patch-clamp recording method (Figure 8, Table 5) [92]. They found that 11-fluorobenzylidene STX (**33c**) showed selective and potent inhibitory activity against Na_V_1.2 (IC_50_ = 7.7 ± 1.6 nM), compared to the other subtypes tested. 11-Benzylidene STX (**33a**) and 11- nitrobenzylidene STX (**33d**) showed potent inhibitory activity against Na_V_1.5, with IC_50_ values of 94.1 ± 12.0 nM and 50.9 ± 7.8 nM, respectively. These compounds are the most potent TTX-r modulators among STX derivatives so far reported, except for ZTX (**8**) [49].

## 3. Stereoselective Synthesis of The Isoxazolidine Moiety of ZTX (8), And Its Introduction at C13 in A Model Compound

As described in the introduction, ZTX (**8**) has a characteristic macrolactam structure from C6 to C11, involving an isoxazolidine ring system. Thus, stereoselective synthesis of the di-substituted isoxazolidine unit in ZTX (**8**) has been examined. In the paper reporting the isolation of **8** in 2004, the amide carbonyl group in ZTX (**8**) at C13 appeared at 156.5 ppm in the ^13^C nuclear magnetic resonance (NMR) spectrum, which is a higher chemical shift compared to other amide carbonyls [49]. This interesting observation might be attributed to the unusual macrolactam structure in ZTX (**8**), and synthetic studies of model compounds have been carried out to understand the origin of this unusual chemical shift. In the following section, we discuss the stereoselective isoxazolidine syntheses reported by Nishikawa’s [93] and Lopper’s groups [94]. 

### 3.1. Synthesis of The Isoxazolidine Part of Zetekitoxin (8) from D-ribose by Nishikawa And Co-workers

In 2009, Nishikawa and co-workers reported the stereoselective synthesis of isoxazolidine **42** from D-ribose (**34**) (Scheme 5) [93]. They firstly synthesized nitroolefin **36** from aldehyde **35**, which was derived from D-ribose (**34**) by means of a Henry reaction followed by dehydration with mesylation. After reduction of the double bond in **36** with NaBH_4_, the resulting nitroalkane **37** was treated with Boc_2_O to produce dihydrooxazole **39a** and **39b** with 86% yield, as a diastereomeric mixture at C16 in a ratio of 3:1. In this reaction, nitrile oxide **38** was generated first, and a 1,3-dipolar cyclization reaction occurred simultaneously. The major transition state model is shown in Scheme 5. The major diastereomer **39a** was reduced stereoselectively with NaBH_3_CN to isoxazolidine **40**. After acetylation of the amine in **40**, isoxazolidine **42**, which has the same stereochemistry as ZTX at C15 and C16, was obtained in three steps: (1) deprotection of acetonide with TFA, (2) oxidative cleavage of diol with NaIO_4_, and (3) reduction of the resulting aldehyde with NaBH_4_. 

### 3.2. Stereoselective Synthesis of The Isoxazolidine Part from Methyl α-d-glucopyranoside by Lopper and Co-Workers

In 2015, Lopper and co-workers reported a synthesis of isoxazolidine **59** (Scheme 6) [94]. Aldehyde **45** was synthesized from commercially available methyl α-d-glucopyranoside (**43**) by the iodination of **44** with iodine and PPh_3_, acetylation of the hydroxyl group, and reductive cleavage of the pyran ring with zinc in acetic acid [95]. Then, intramolecular 1,3-dipolar reaction of the terminal olefin with nitrone, which was generated from aldehyde **45** by reaction with hydroxylamine **46**, took place stereoselectively to afford **48** via **47** with 52% yield. After deprotection of acetate in **48** with sodium methoxide [96,97], the resulting triol **49** was treated with NaOI_4_ followed by LiAlH_4_ to give diol **51** with 70% yield in two steps [98,99]. The isoxazolidine synthon **59** in ZTX (**8**) was synthesized from diol **51** in seven steps by selective functionalization of the two hydroxyl groups, followed by *N*-acylation. 

### 3.3. Comparison of the Chemical Shift at C13 in Zetekitoxin (8) with Those in Some Synthetic Models

As discussed above, the ^13^C NMR chemical shift of the carbonyl group at C13 in ZTX (**8**) has been observed at 156.5 ppm [49], which is a higher value compared with usual amide carbonyl groups (170–175 ppm). To address the issue, Nishikawa’s and Looper’s groups independently examined the ^13^C chemical shifts of the carbonyl group at C13 in some model compounds (Figure 9) [93,94]. Simple N-acyl isoxazolidine models **60**, **42**, and **59** showed chemical shifts of 171.0, 172.7, 171.0 ppm, respectively, which are quite similar to those of regular cyclic *N*-acyl amides. However, model compounds **61**–**63** bearing alpha-guanidinoacetyl amide groups showed chemical shifts of 166.0, 168.3, and 167.0 ppm, respectively, being shifted ca. 5 ppm upfield compared to the other simple models.

Nagasawa and co-workers examined the chemical shift at C13 of **70**, which has an STX skeleton; its synthesis is depicted in Scheme 7 [100]. They firstly aimed to obtain carboxylic acid **66** from alcohol **64** by oxidation. They examined various oxidants and conditions, but it appeared that the hydroxyl group in **64** was unreactive due to its axial orientation, and no reaction occurred, or unexpected side reactions proceeded. Finally, they found that 2-azaadamantane *N*-oxyl (AZADO)–NaClO and NaClO_2_ [101,102] were effective, resulting in carboxylic acid **66**, which was obtained with 79% yield after TMSCHN_2_ treatment of the crude carboxylic acid **66** to hydrolyze the methyl ester **67**. Condensation of carboxylic acid **66** with isoxazolidine **40 [93]** in the presence of 4-(4, 6-dimethoxy-1,3,5-triazin-2-yl)-4-methylmorpholinium chloride (DMT-MM,**68**) [103], followed by deprotection of the Boc group and acetal with TFA, gave amide **70** in 98% yield. Unfortunately, the chemical shift of the carbonyl group in **70** was observed at 166.1 ppm, slightly higher than that of **62** or **63**, but still lower than that of ZTX (**8**). The chemical shift in ZTX (**8**) may reflect the characteristic spatial structure associated with the presence of the macrolactam moiety. 

## 4. Synthesis of The Characteristic Macrocyclic Structure of ZTX (8) by Looper’s Group

Looper and co-workers have reported macrocyclic compound **72** as a model for ZTX (**8**) (Scheme 8) [71]. After deprotection of the TBDPS ether group at C13 with TBAF, the resulting alcohol was reacted with iodoacetic acid in the presence of 1-(3-dimethylaminopropyl)-3-ethylcarbodiimide (EDC) and N,N-dimethyl-4-aminopyridine (DMAP) to give iodoester **71** with 58% yield. When iodoester **71** was treated with a strong base, *tert*-butylimino-tri(pyrrolidino)phosphorane (BTPP), intramolecular alkylation proceeded at C11, and the corresponding macrolactone **72** was obtained in 48% yield. It should be possible to construct the macrolactam structure of **8** via a similar strategy, and this should also resolve the chemical shift issue in ZTX (**8**). 

## 5. Conclusions

Here, we have reviewed recent progress towards the total synthesis of zetekitoxin AB (**8**, ZTX). Although this goal still remains elusive, there have been some significant synthetic advances in the construction of characteristic structures of ZTX, such as (i) the carbon–carbon bond at C11 in the STX structure, (ii) stereoselective construction of the substituted isoxazolidine moiety at C15 and C16, and (iii) the macrocyclic structure from C6 to C11. Since ZTX has potent inhibitory activity, even towards tetrodotoxin-resistant (TTX-r) Na_V_s, a total synthesis of ZTX and its analogs is expected to provide useful tools for chemical biological studies of Na_V_s, overcoming the severely restricted availability of natural ZTX.

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
