# Peer review of "Synthetic Approaches to Zetekitoxin AB, a Potent Voltage-Gated Sodium Channel Inhibitor"

_marinedrugs, 2019, doi:10.3390/md18010024_

Round 1
Reviewer 1 Report
Adachi and colleagues provide a comprehensive overview on the state of the art on synthesis of zetekitoxin AB. This molecule is of particular interest because of its potent inhibition of voltage-gated sodium channels. However, successful synthesize of this complex molecule remains a challenge. I only have a few minor remarks that should be addressed:
In the text, both Nav and NaV are used. This should be uniform throughout the manuscript.
Figure 2: Did the authors construct the model of STX with Nav1.7? If yes, the methodology needs to be described. If not, the appropriate reference needs to be provided
Line 85 & 86: why are these references marked in yellow?
The reference list needs to be corrected according to the guidelines of Marine Drugs. Now, several reference numbers contain more than 1 reference and sometimes the reference seem to be incomplete (for example reference 14, line 419)
Author Response
Response to Referee #1:
[1] In the text, both Nav and NaV are used. This should be uniform throughout the manuscript.
Our response:
We appreciated the comment. We use “NaV” in the revised manuscript.
[2] Figure 2: Did the authors construct the model of STX with Nav1.7? If yes, the methodology needs to be described. If not, the appropriate reference needs to be provided
Our response:
Thank you for the valuable suggestions. We have not constructed the model. The model is described in the Reference [45] (original manuscript: reference [14]), and we added the reference [45] at the caption in Figure 2 in the revised manuscript.
Page, 3 line 2
Original
Figure 2. A: Top view of the structure of STX-NaV1.7 complex. B: Specific interactions in STX-NaV1.7 complex.
Page, 3 line 2
Revised
Figure 2. A: Top view of the structure of STX-NaV1.7 complex. B: Specific interactions in STX-NaV1.7 complex [45].
[3] Line 85 & 86: why are these references marked in yellow?
Our response:
We appreciated the comment. As reviewer pointed out, this was our mistake. We removed the yellow mark for references in the revised manuscript.
[4] The reference list needs to be corrected according to the guidelines of Marine Drugs. Now, several reference numbers contain more than 1 reference and sometimes the reference seem to be incomplete (for example reference 14, line 419)
Our response:
We appreciate the comments, and we really apologize about many errors in the manuscripts. As reviewer pointed out, we revised all references in accordance with the guidelines of Marine Drugs.
Reviewer 2 Report
This is an interesting review but must be improved.
Authors should uniform abbreviations in all manuscript. In page 1 we could find Nav or NaV or NaVCh. Besides they must identify abbreviations with the full name, at least the first time it appears. Example in Figure 3 GTX (gonyautoxin).
Page 2 lines 37-39. Please review the origin of STX and clarify that dinoflagellates synthesize the toxin.
Authors have to use the correct name of the toxins. For instance “gonyautoxins” instead of “goniotoxins”. Page 3 line 67
The subtitles must be in italics but no bold font. Page 3 line 63
Authors must present the figures appropriately with a suitable figure caption.
In Page 6 Figure 6 should be divided into 3 (a, b and c): a) Isomerization of a-SEA to b-SEA b) Inhibitory activity of SEA (authors must include the Nav isoform studied in the cell-based assay with Neuro 2A cells) c)Structure of GTX III and IC50 (Nav 1.4).
Page 11 Tables 4 and 5 Should be divided in a figures and tables.
Author Response
We attached our response to the reviewer 2.
